# Molecular Dynamics and Docking Simulations of Homologous RsmE Methyltransferases Hints at a General Mechanism for Substrate Release upon Uridine Methylation on 16S rRNA

**DOI:** 10.3390/ijms242316722

**Published:** 2023-11-24

**Authors:** Aaron Hernández-Cid, Jorge Lozano-Aponte, Thomas Scior

**Affiliations:** 1Biochemistry Department, BioPlaster Research Institute, Puebla C.P. 72260, Mexico; aaron@bioplaster-inc.com; 2Escuela de Ingeniería y Ciencia, Instituto Tecnológico y de Estudios Superiores de Monterrey, Campus Puebla, Puebla C.P. 72453, Mexico; jorge.lozanoaponte@tec.mx; 3Departmento de Farmacia, Facultad de Ciencias Químicas, Ciudad Univeristaria, Benemérita Universidad Autónoma de Puebla, Puebla C.P. 72570, Mexico

**Keywords:** methyltransferase, 16S rRNA methylation, RsmE, molecular dynamics, antimicrobial resistance

## Abstract

In this study, molecular dynamics (MD) and docking simulations were carried out on the crystal structure of Neisseria *Gonorrhoeae* RsmE aiming at free energy of binding estimation (ΔG_binding_) of the methyl transfer substrate S-adenosylmethionine (SAM), as well as its homocysteine precursor S-adenosylhomocysteine (SAH). The mechanistic insight gained was generalized in view of existing homology to two other crystal structures of RsmE from *Escherichia coli* and *Aquifex aeolicus*. As a proof of concept, the crystal poses of SAM and SAH were reproduced reflecting a more general pattern of molecular interaction for bacterial RsmEs. Our results suggest that a distinct set of conserved residues on loop segments between β12, α6, and Met169 are interacting with SAM and SAH across these bacterial methyltransferases. Comparing molecular movements over time (MD trajectories) between *Neisseria gonorrhoeae* RsmE alone or in the presence of SAH revealed a hitherto unknown gatekeeper mechanism by two isoleucine residues, Ile171 and Ile219. The proposed gating allows switching from an open to a closed state, mimicking a double latch lock. Additionally, two key residues, Arg221 and Thr222, were identified to assist the exit mechanism of SAH, which could not be observed in the crystal structures. To the best of our knowledge, this study describes for the first time a general catalytic mechanism of bacterial RsmE on theoretical ground.

## 1. Introduction

Ribosomal RNA small subunit methyltransferase E (RsmE) is a methyltransferase (MTase) and was first discovered in *Escherichia coli* (*E. coli*) [1]. It transfers methyl groups from methyl group donor S-adenosylmethionine (SAM) to uridine U1498 in the bacterial 16S ribosomal RNA (rRNA) [2]. In contrast to protein coding messenger RNA (mRNA for gene expression), rRNA, along with transfer RNA (tRNA), are noncoding yet mRNA and tRNA are directly participating in protein biosynthesis and rRNA indirectly regulates through methylation. Molecular mechanisms of rRNA methylation are highly conserved and span all kingdoms of life [3]. In bacteria, it plays a major role in the regulation of ribosome activity and the local fine-tuning of rRNA structure, which is of epidemiological importance because it is a mechanism of antibiotic resistance [4] and epigenetic regulation of protein synthesis [5]. Nineteen members of the MTase family are responsible for the catalysis of post-transcriptional modifications of adenine, guanosine and cytosine but only one type (RsmE) modifies uridine in the 16S rRNA and three in the 23S rRNA [6].

At an atomic scale, the methylation target of RsmE in *Neisseria gonorrhoeae* (*N. gonorrhoeae*) constitutes the nitrogen atom at position N3 on uridine U1497 of the 16S rRNA. Methyl transfer produces m^3^U1497 as well as S-adenosylhomocysteine (SAH) as a byproduct. Primary sequence comparison with *E. coli* demonstrates that its U1497 corresponds to U1498 in *E. coli* and belongs to the highly conserved A-site, which is significantly important for protein biosynthesis [1], as well as for the interaction with the ribose-phosphate backbone of the codon region [7]. A schematic drawing (Figure 1) shows the relative position in the ribosomal context where RsmE catalyzes the methylation of uridine (U). The physiological role of U1498 and its methylation to m^3^U1498 were confirmed in experiments of *rsme* knockout mutants that showed lagged growth rates compared to those of the wild-type strains [1]. Studies in *Mycobacterium tuberculosis* have shown that the methylation of U1498 has an indirect effect on the binding of aminoglycosides and, hence, it inherently acts as an effective mechanism of drug resistance [8].

In a seminal work, RNA methylation was embedded in a wider biochemical context of posttranslational processing, gene expression and regulatory mechanism thereof in the field of epitranscriptome [9]. Clinical relevance has the uridine methylation in light of the recent developments of mRNA vaccines for humans [10]. Other types of methylation exist, for instance concerning oxygen atoms on nucleotides (nt) in RNA, but they require different mechanisms of action [11].

Sequence and structure analyses of published RsmE data indicated that RsmE belongs to the SPOUT (SpoU-TrmD) superfamily, which has been conserved not only in bacteria but also in eukaryotes. SPOUT members—including RsmE—form dimers that are essential for the catalytic activity. They share a characteristic SPOUT domain with highly conserved primary sequences (homology), and shared protein functions based on two common structural features: (i) a SAM-binding fold as a three-layered ɑ/β fold with a central β-sheet of 5–6 strands surrounded by ɑ-helices on each side; as well as (ii) a PUA-like domain in the N-terminal segment [2,12].

In addition to RNA rearrangements including rRNA or tRNA maturation, chain cuts and splicing or chaperone folding, RNA bases alone can be modified. In our context, transmethylation of a nt requires an initial nucleophilic attack to the methyl carbon atom which in turn must be activated by an adjacent positive charge. Here, it is a sulfonium cation (a sulfur atom with a total charge of +1) on SAM. As a methyl group donor, the methyl group attached to its sulfur atom transferred to the attacking nucleophile after its accommodation in the cavity of the active (catalytic) site of the methyl transferase enzyme. Nucleophilic moieties contain electronegative atoms like N, O and S, or negatively charged groups such as carboxylate, phosphate (esters) and enolate anions. Here, the nucleophilic center to be methylated is the nitrogen atom in position 3 of uridine at the nt sequence position 1498 (m^3^U1498). Of note, compared to our target enzyme for N-methylation, C-methylation requires a totally different mechanism for m^3^U1498.

A deeper understanding of the biochemical steps during the catalytic methylation cycle of RsmE is still lacking. Hence, we propose a computational approach to analyze the exit mechanism of SAH by molecular dynamics simulations [13,14,15,16].

We carried out molecular dynamics (MD) simulations of *N. gonorrhoeae* RsmE and, after inspection of two other species, *E. coli* and *Aquifex aeolicus* (*A. aeolicus*), we propose a more general mechanism for bacterial RsmE. To the best of our knowledge, this is the first MD analysis on this matter to date. The outcome suggests, on theoretical grounds, a common bacterial mechanism that includes the following features: (i) a pair of Arg/Gly for the entry of SAM, (ii) a pair of Arg/Thr to stabilize SAM during catalysis by arginines and glutamates, and (iii) a pair of isoleucines for SAH exit.

## 2. Results

### 2.1. Sequence Alignment and Structural Superposition of the Bacterial RsmEs

The structural alignment of RsmE homologs (Figure 2) revealed close structural similarity, except for *Thermus thermophilus,* which diverged significantly from the typical α/β architecture of the PUA fold, which generally comprises a six-stranded pseudo-β-barrel capped by an α-helix on each apical side of the pseudobarrel (see Appendix A in the Appendix A, for a more detailed analysis of the RsmE structure). The *T. thermophilus* from the PUA-like domain at the N-terminus is likely to be adaptive to the recognition of target RNA structures with higher stability [17].

The C-terminus of bacterial RsmE embraces a Rossmanoid fold with conserved motifs harboring functional residues for (i) SAM binding, (ii) methyltransferase activities, and (iii) a dimerization interface. The dimeric nature of 5VM8 was demonstrated in the asymmetric unit of the crystal, which shows a dimer and displays a set of conserved amino acids that form saline bridges to build an area of approximately 1408 Å^2^ (see Appendix A in the Appendix A). A similar network of hydrogen bond interactions was observed in each homodimer, i.e., Glu98 with Arg218, Asp239 with Lys176, Arg218 with Asn71, and a high number of contacts that enforce the interface interaction. Other studies have suggested that RsmE catalysis is carried out by the dimer rather than the monomer [18].

Following the fundamental tenet that highly conserved structures reflect similar protein functions over eons of biological evolution, our MD production runs together with our structural superposition of CTD suggesting that all RsmEs effectively do share closely related catalytic mechanisms (Figure 2). The aforementioned Rossmanoid fold in the C-terminus is encountered in a plethora of methyltransferases, as it is considered a remnant of a primogenial ancestor spread across many protein families [19]. The central seven-stranded parallel β-sheet (β6-β12) is sandwiched between two layers of helices on either side α4 and α5 on one side, and α2, α3 and α6 on the other, and a deep trefoil knot is formed by threading the β11-β4 segment C-terminus (Figure 3 and Figure 4). 

Comparison between crystal structures and primary sequences of RsmE proteins revealed a higher degree of structural conservation over sequence. *E. coli* and *A. aeolicus* RsmE are homologous, with 39.33% and 29.86% identity, respectively, with respect to the *N. gonorrhoeae* sequence. The structural similarity was evaluated by measuring the root mean square deviation (RMSD), obtaining a score of 1.39 Å over 868 backbone atoms for *E. coli*, and 1.37 Å over 708 backbone atoms for *A. aeolicus*, compared with a total number of 943 backbone atoms of *N. gonorrhoeae* crystal structure as reference [20] (Figure 2, see Appendix A in the Appendix A). All the RMSD values from the RsmE crystal structure structural comparison are presented in Appendix A in the Appendix A.

Although *E. coli* RsmE displays a high degree of structural similarity and identity with *N. gonorrhoeae*, analysis of the crystal structures combined with our MD results suggest that it is even more closely related to *A. aeolicus*. To further investigate the catalytic mechanism of RsmE, the structural model of 5VM8 was taken as a reference because it yielded the most consistent results and unveiled the most continuous electron density map. *N. gonorrhoeae* RsmE overlapped on its electron density map and the ligand model on a Polder map. In particular, the underlying electron density map was obtained from a separate omit map file called 2Fo-Fc. It has been made available for research in the structural biology field along with the corresponding PDB entry 5VM8. As a most valuable asset, 2F0-Fc omit maps offer information regarding how accurately the model accounts for the experimental data. The omit map was fetched from the PDB server and visualization that was carried out under PyMol. Both consistently agree with the observed data, and only a small segment at the beginning of the NTD lies outside the electron density map (Figure 4); the same applies to the final part of the CTD. The SAM model fits perfectly when water molecules or other solvent molecules are excluded. At this stage, MD simulations proceeded to determine the enzyme–substrate contacts and estimate the binding energies.

In the first line of the MSA study (Figure 3) appears the reference sequence of *N. gonorrhoeae* RsmE. Its 3D structure was under scrutiny by MD, docking and inspection. Its residues were highlighted in yellow (resp., green) to indicate the computationally observed interaction with SAM and SAH (resp., only SAM) by docking and DS-ViewerPro (resp., by crystal structure inspection with PDB-NGL viewer). Interactions seen with both molecular model display editors are shaded in gray color. Interactions exclusively seen by MD are shaded in light purple. Ile171 and Ile219 are shaded in light pink. Conserved positions are depicted in red letters and non-conserved residues are in orange letters. Next, all homolog residues in Lines 2 through 11 were colored identically. The different RsmE motifs (see Appendix A, in the Appendix A) are marked in red (α-helices) and light blue (β-strands) in the *N. gonorrhoeae* sequence.

The reference sequence of *Neisseria gonorrhoeae* RsmE were compared to the following sequences from PDB entries: 5VM8 (unpublished PDB entry but mentioned in the publication of Pinotsis and Waksman from 2017 [18]); 4E8B for *E. coli* by Zhang et al. from 2012 [2]; 2EGW for *A. aeolicus* (unpublished PDB entry but mentioned in Kumar et al., 2014 [12]; 1NXZ for *Haemophilus influenzae* [21]; 5O95 for Legionella pneumophilia also by Pinotsis and Waskman in 2017 [18]; 1V6Z for *Thermus thermophilus* (unpublished PDB entry but mentioned in Basturea et al., 2006 [1]); 3KW2 for *Porphyromonas gingivalis* (unpublished PDB entry but mentioned in Kumar et al., 2014 [12]); 1VHK for *Bacillus subtilis* was registered in PDB by Badger et al. in 2005 [22]; 4L69 for *Mycobacterium tuberculosis* by Kumar et al. in 2014 [12]; 1Z85 was deposited in 2005 for *Thermotoga maritima* (unpublished PDB entry but mentioned in Zhang et al., 2012 [2]); and 4J3C was released in 2013 for Sinorhizobium melioloti (also an unpublished PDB entry). 

### 2.2. Interaction Patterns of SAM and SAH in the RsmE Binding Site of N. gonorrhoeae from the Molecular Dockings and the Crystal Structure

The crystal structure of RsmE in complex with SAM/SAH was analyzed along with the set of interactions involved (Figure 5). The SAM methionine tail can be broken down into three chemical substructures: the carboxyl, amine, and methyl sulfide groups. The carboxyl group of SAM faces towards the binding site exit without making any other contact with RsmE, whereas its amino group is forming H-bond with the backbone oxygen atom of the peptide bond on Gly195. Both chemical groups lie at a distance from the methyl group that is exposed to the RNA contact site. Methyl sulfide is positioned next to the H-bond acceptor region, which is highly polar compared to the major hydrophobic character of the binding site. The binding site hydrophobic regions certainly exert weak interaction forces with the residue backbones to the methylated RNA *in statu nascendi* and facilitate SAH to leave.

Concerning the adenine ring, shape compatibility exists between the hydrophobic receptor wall and H-bond acceptor residues that stabilize the positioning of the SAM or SAH. These interactions are consistent with those observed in the crystal structure and are supported by the calculated docked conformations. Indeed, the SAM tail maneuvers in sufficient space for free movement during RsmE conformational changes before it is catalyzed to homocysteine (Figure 5A).

The observed active conformation of SAM from the RsmE crystal complex was superimposed with the docked conformations of SAM and SAH. SAM of the crystal complex and the final docked solutions (FDS) showed practically the same atom-by-atom contact patterns at the RsmE binding site (Figure 5B). The backbone oxygen atoms of Met169, Gly195, Leu215, Gly216 and Arg218, along with the backbone nitrogen atoms of Gly192, Leu215 and Leu220, form hydrogen-bonds with SAM. Intriguingly, they could be observed only in the MD playback but not in the crystal structure itself. In contrast to the Gly192 interaction, which was observed only in the crystal structure of SAM. For RsmE-like homolog from *Mycobacterium tuberculosis*, it has been reported that the equivalent Arg218 and its neighboring leucine and glycine residues are essential for methylation catalysis. Conserved arginine and hydrophobic amino acids were rearranged to interact with the SAM adenine ring and U1498 of p44 of *E. coli* [12], suggesting that the structural basis of bacterial RsmE evolved to recognize a specific rRNA sequence. Indeed, RsmE associates with the RNA-binding site in addition to a variety of rRNA sequences that carry the conserved A(N)GGAX motif [5]. At the time of finalizing this study, a crystal structure of PUA-like domain in complex with p44 and SAM/SAH together has been made available (last visit, June 2023).

Docked SAH exhibited the same interaction pattern as SAM for its adenine ring with almost the same residues, except for the backbone nitrogen of Gly192. Instead, the carboxyl and amine groups on SAH contacted the side chain nitrogen atoms of Arg111 and the backbone oxygen of Glu194, respectively. This interaction dissimilarity between SAM and SAH side chains is due to the inward displacement of the sulfur atom that allows the methyl group to shift over during the ongoing methyl transfer process, as displayed in Figure 5A.

The sulfur atom displacement (or shift) of the three ligands can be fully understood when considering the following facts: two ligands can be lumped together since the sulfide S-atom is methylated, becoming a cation. They are accommodated in the binding cleft in the same way, because the surrounding spatial requirements of the cavity are the same. In stark contrast, the S-unmethylated ligand is smaller and approaches more deeply into the occupation zone at the binding site. Since the sulfur atom sits in the middle of a sidechain on the scaffold, it becomes evident that the binding conformations vary from one to the other group. For better viewing, we extracted the active conformations (Figure 5). Our molecular dynamics simulations reflect this side chain displacement (or shift) at room temperature.

The pivotal role of Gly192 in recognizing SAM or SAH is evolutionarily determined. Gly192 belongs to the conserved motif GPEGX, located on the loop between β11 and α5 of *N. gonorrhoeae* RsmE. Of note, the aforementioned differences in the SAM and SAH interaction patterns do not affect affinity since their binding energies were computed to have similar values.

### 2.3. The Bacterial RsmE Binding Site Is a Crevice with a Network of Hydrogen Bonds and Hydrophobic Interactions

The *E. coli* RsmE monomer was simulated for 10 ns by MS using the same protocol described in the simulation section for *N. gonorrhoeae* RsmE. However, *E. coli* RsmE was energetically minimized before obtaining a relaxed binding site with a shape similar to that observed in *N. gonorrhoeae* and *A. aeolicus* (for details, see also Appendix A in the Appendix A).

The crystal structures of SAM/SAH in complexes with *N. gonorrhoeae* and *A. aeolicus* showed that their estimated binding energy values were very close to each other, but they were highly different from the docked SAM and SAH within the *E. coli* RsmE binding site (Figure 6A–C).

While the ligand binding energies in *N. gonorrhoeae* and *A. aeolicus* are approximately −9 kcal/mol, the mean binding energy value of SAM and SAH in *E. coli* RsmE is approximately −7 kcal/mol, i.e., a 30-fold affinity reduction (Appendix A in Appendix A). The binding sites of the three species were found very similar to those of the crevice. Close to the amine head group of the adenine ring, a distinct polar amino acid (Arg218 in *N. gonorrhoeae*, Arg220 in *E. coli*, or Tyr221 in *A. aeolicus*, resp.) was positioned to accept an H-bond that helps fix the ligand at the binding site (Figure 6D–F). This polar amino acid is positioned at the edge of a highly hydrophobic hole, which enhances ring stabilization. At the other extreme, the aperture size also varies depending on the species and works as a hand to guide the SAM/SAH tails using H-bonds and hydrophobic interactions (Figure 6G–I).

In Figure 6A–C, three final docked poses for SAM and SAH are displayed to analyse their computed interaction pattern with the RsmE of *N. gonorrhoeae*, *A. aeolicus* and *E. coli*, respectively. In each case, they show almost identical positions when compared to the x-ray conformations of SAM and SAH with *N. gonorrhoeae*: 5VM8 and *A. aeolicus*: 2EGW, respectively (Figure 6D–F), including those obtained with the RsmE of *E. coli*, the source of which constitutes an unliganded crystal structure. Prior to docking with RsmE of *E. coli*, its crystal structure was prepared by geometry relaxation through MD. This way, strain energy lying on the tighter apo-form of the protein structure was released to simulate assumed induced fit processes between ligands and receptors upon binding [23]. Next, the resulting docked poses for SAM as well as SAH were compared to their original crystal structures. For not being back docking results, it could be assumed that poses were not found identical but at least in acceptable keeping with the overall positions. The adenosine rings were found in close proximity to the back docked poses after superposing them. The latter comes as no surprise as the crystal structure ligand conformations highly resemble those of the adenosine and sugar rings (Figure 6G–I). It is not farfetched to conclude that binding SAM or releasing SAH share highly similar mechanisms among bacterial organisms.

In particular, the aperture next to the adenine ring stays widely open in *N. gonorrhoeae* and *A. aeolicus* compared to *E. coli* (Figure 6A–C), which exhibited a narrowly closed hole. These observations are consistent with those seen in *N. gonorrhoeae* and *A. aeolicus* crystal structures. The final poses of SAH by back docking in addition to its docked crystal pose were found to be very similar to that of SAM in its crystal complex. Precisely, the sulfur atom can be taken as a reference point. It occupies the same spot with a tiny distance variation regarding the atoms of neighboring backbone residues (Figure 6I). Finally, the *A. aeolicus* RsmE-binding site had more regions with H-bond donor residues among the three species. This high number does not seem to have a major drawback in the conformation that these ligands adopt when superimposed with the *N. gonorrhoeae* crystal structure (Figure 6).

More details concerning docked poses and resulting binding energies and interaction patterns observed by inspecting the crystal structure and molecular docking are documented in Appendix A in addition to Appendix A. Across the three RsmE species, a pair of one polar and non-polar residues are either conserved or replaced during evolutionary times by others with equivalent chemical groups. In particular, the following three pairs were found which assist SAM and SAH in entering (entrance) and leaving (exit): (i) Leu215/Arg218 for *N. gonorrhoeae*, (ii) Leu208/Tyr211 for *A. aeolicus* and (iii) Leu217/Arg220 for *E. coli*. In all cases, the polar amino acid comes in close proximity to the amine group of the adenine ring and guides it around a hydrophobic crevice made of leucine side chains. Moreover, Leu220 also plays a fundamental role in the stabilization of the adenine ring in *N. gonorrhoeae* (Figure 6D–F). The equivalent residues in *A. aeolicus* (Leu213) and *E. coli* (Leu222) appear to share the same function. Likewise, the interaction of sugar hydroxyl oxygen (O2’) with Met169 in *N. gonorrhoeae* is analogous in *A. aeolicus* with Leu161 and Leu 171 in *E. coli*. Glu194 and Glu187 from *N. gonorrhoeae* and *A. aeolicus,* respectively, may contribute to stabilizing SAH exit by guiding the homocysteine tail (Figure 6G,H).

### 2.4. The Dynamics inside RsmE during SAH Release

In this study, the CHARMM force field was selected because it includes the MD parameters for drug-like molecules containing sulfur [24,25], which in turn allows the analysis of the interaction of SAM/SAH with RsmE. Molecular dynamics analysis was carried out with the RsmE monomers of *N. gonorrhoeae* alone (5VM8m) and in the presence of SAH (5VM8m + SAH). The backbone of 5VM8m and 5VM8m + SAH conventionally remain stable near 2 Å, RSMD_5VM8m_ = 2.73 Å and RSMD_5VM8m+SAH_ = 2.89 Å, with similar fluctuations over time. This is consistent with other MD studies with methyltransferases in complex with nanaomycin and hydralazine as inhibitors [26,27] and reports of folded RsmE that showed a RMSD of 2.5 Å (Figure 7A) [28].

The average RMSD values indicate that the backbone atoms experience a similar deviation in the absence and presence of SAH, and hence RsmE does not undergo unfolding during methyl transfer. In agreement with backbone folding, the Rg graphs and mean values indicate that the general form of the RsmE monomer is constant (Figure 7B) [29]. The SAH trajectory within RsmE and the dynamics related to its exit mechanism after RNA methylation are displayed in Figure 8.

The distance between the SAH adenine ring and the amine group with Leu215, Arg218 and Leu220 residues remained practically constant at 3 Å for the entire simulation (Figure 9), maintaining this part of the molecule very close to the crystal pose, further reinforcing the importance of the role of the polar amino acid. The interaction distance with the Met169 residue does not show the same stability as the residues, since it is kept 3 Å away from one of the hydroxyl groups (O2’) of the SAH sugar ring (Figure 9A). Nevertheless, the interaction distance between the same hydroxyl group (O2’) and the Gly192 residue increases as the simulation time progresses, indicating that the sugar ring undergoes a major position change as a function of time compared to the adenine ring. 

### 2.5. SAH Exchange between Arg221 and Thr222 Is Based on Swinging Side Chains

One hallmark of molecular dynamics is the high mobility of the homocysteine SAH tail attenuated when its carboxyl group was exchanged between the side chain of Arg221 and the backbone of Thr222. An abrupt dislocation or leap occurs which is up to three times larger when compared distance analysis (Figure 10E–H or Figure 11B). In Figure 11B,C, the conformations of side chains of Arg221 and Thr222 changed randomly when SAH was absent. In contrast, however, when it is present, its side chain terminal carboxyl group adopts an orderly and defined back-and-forth rotation similar to a swing. MD analysis detected this differential behavior. In particular, atoms O5 and O6 of SAH strongly kept distances to the aforementioned residues remarkably constant over time. In line with this unique behavior (distance pattern), atom O2 of the SAH hydroxyl group gradually weakened its contact with Gly192, suggesting that the displacement of SAH was composed of two synchronous events: (i) the stable interaction with Arg221/Thr222 in concert with the contact loss to backbone atoms of Gly192. In other words, the smaller the distance between SAH and Thr222, the stronger the separation to Gly192 (Figure 10A,G,H). These observations are observed in neither experimentally observed nor docked poses.

### 2.6. An Isoleucine Gate Is Active in Concert with Arg/Thr Contacts to Release SAH

The release of SAH as a leaving moiety is mediated by two isoleucine residues acting as a gate and programmed as a preset mechanism. When the distance between the delta carbons (Cδ) of Ile171 and Ile219 is less than or approximately 6 Å, the binding site adopts a closed conformation, similar to a double latch made of two opposite rods (Figure 8 at 0 ns, and Figure 11D). In the RsmE binding site, the two isoleucine residues face each other, aligned with the long carbon chain of one to the short carbon chain of the other, and form a temporary pipe-like pocket. This is mostly observed in the presence of SAH, which is consistent with what is observed in the crystal structure of RsmE in *N. gonorrhoeae* (Figure 4). In contrast to the closed conformation, the delta carbons (Cδ) of both isoleucines were separated slightly above 6 Å, and the binding site was widely open for a short period of time (Figure 11F).

These data, together with the crystal structure and docked studies, construct a chain of successive interactions. Initially, SAM was guided inside the binding site with the help of the polar amino acids Arg218 and Met169 and the non-polar Leu215 and Leu220. Lastly, the positioning of its tail comes into play by Gly192, after which the SAH exchange between Arg221 and Thr222 occurs prior to promoting hydrophobic interactions with the double isoleucine gate (Ile171 and Ile219), keeping the latches close until rRNA methylation is accomplished.

The general protein shape did not undergo significant conformational changes (Figure 7). However, when the RMSD and radius of gyration on different residues and atoms of the RsmE binding site were plotted and measured, they showed clear differences in the alpha carbons, suggesting that the binding site undergoes significant shape variation in the absence of SAH (Figure 12 and Figure 13).

## 3. Discussion 

### 3.1. Literature-Based Evidence and Molecular Dynamics Results 

Our computed results advance towards our understanding of the general methylation transfer among bacterial RsmEs. *N. gonorrhoeae* RsmE exhibits dynamic folding as methylation advances and sets some fundamental amino acids for every step, which is consistent with other studies [2,5,12]. Heng et al. demonstrated that a set of leucine and other polar amino acids from the binding site are essential for catalytic activity. These amino acids were regarded in this study as part of the whole system and agreed to the same residues proposed here. Additionally, SAXS measurements of *E. coli* RsmE confirmed our observations of RsmE folding and the major conformational changes that occur during methyl transfer [2]. 

### 3.2. Structural Insight and Bacterial Resistance 

The resolution of a ternary complex of 16S rRNA, SAM and RsmE could pave the way towards a deeper understanding concerning the atomic intimacy of RNA methylation in bacteria. In more general terms, each contribution to further understanding RsmE catalysis will be a step forward for the design of prospective inhibitors to reverse the bacterial resistance against aminoglycoside antibiotics.

## 4. Materials and Methods

### 4.1. RsmE Multiple Sequence Alignment and Molecular Structure Superposition

Eleven RsmE UniProt sequences from eleven species along with their structural data from RCSB PDB (see Appendix A) were compared. To this end multiple sequence alignments (MSA, see Appendix A) were carried out on the Clustal Omega server [30,31,32,33]. Monomeric crystal structures of RsmE were superimposed by SPDBV using default settings (Figure 14) [34]. In all cases, the *N. gonorrhoeae* sequence as well as its 3D structure were taken as reference.

### 4.2. Molecular Design for Computational Simulations

Monomeric 3D models of RsmE from *N. gonorrhoeae*, *E. coli* and *A. aeolicus* were extracted from crystal complexes (PDB codes: 5VM8, 4E8B and 2EGW). While the ligand structures of SAM and SAH were extracted from liganded complexes (PDB codes: 5VM8 and 2EGW). Their 3D models were prepared under SPDBV (Appendix A) [34]. In solution, the methionine scaffold remains in twitter-ionic form—likewise for any amino acid in pure water—while the adenosyl side chain remains unprotonated for an estimated 99 to 99.9% (pKa 4–pH 7 = 99.9% free base). In stark contrast, hydrophobic environments hamper dissociation (cf. Section 4.5). Of note, the drug bank entries for SAM and SAH display wrong ionization states, albeit the overall total charges are correctly assigned to them: monocationic SAM and neutral SAH (https://go.drugbank.com/drugs/DB00118 and https://go.drugbank.com/drugs/DB01752; both accessed on 6 September 2023). Respecting the predefined total charges, Gasteiger partial charges were assigned to both ligands using VegaZZ [35]. The chirality of their 3D models was verified according to their stereospecific descriptions (Figure 14). The IUPAC name for SAM is *[(3S)-3-amino-3-carboxypropyl]-[[(2S,3S,4R,5R)-5-(6-aminopurin-9-yl)-3,4-dihydroxyoxolan-2-yl]methyl]-methylsulfanium* [36]. SAH is denominated as *(2S)-2-amino-4-[[(2S,3S,4R,5R)-5-(6-aminopurin-9-yl)-3,4-dihydroxyoxolan-2-yl]methylsulfanyl]butanoic acid* [37]. In the 3D models of proteins, all hydrogen atoms were added under Autodock Tools [38].

### 4.3. Molecular Docking of SAM and SAH at the RsmE Binding Site of N. gonorrhoeae, A. aeolicus and E. coli

Active conformations of SAM and SAH and their receptor affinities (ΔG_binding_) were determined at three different RsmE binding sites of *N. gonorrhoeae, A. aeolicus and E. coli* using Autodock 4.2 [38].

For validation of the docking protocol SAM and SAH were successfully docked back into their respective observed binding positions: SAM into 5VM8 from *N. gonorrhoeae* and SAH into 2EGW from *A. aeolicus* (back docking) before blind docking of both ligands n *E. coli* RsmE (Figure 14). Docking was performed on a grid box centered around each ligand with a box size of 50 × 50 × 50 Å^3^. The genetic and local search algorithm GALS was taken for 256 runs, 2,500,000 evaluations and elitism of 3 (most suited survivors of each run) [39,40]. The resulting docked poses were analyzed after conformational clustering by root mean square distance (RMSD = 2.5 Å with crystal structures of SAM and SAH as references). Docking outcome and ligand–receptor interactions were analyzed with AutoDock Tools and Discovery Studio (DS Viewer Pro) [38,41]. Six docked poses were obtained (2 ligands by 3 receptors gave six runs with 2 BkD plus 4 BdD settings) and the selection criterion was the (shortest possible) distance to the reference crystal pose. The selection was also in keeping with two other aspects of hit selection: top-ranked scoring or one of the most populated RMSD clusters. 

### 4.4. Molecular Dynamics of Monomeric N. gonorrhoeae RsmE Alone and with SAH 

The CGenFF server and Python cgenff script from the MacKerell website were used to generate the CHARMM36 force field topology of the ligand and its atom coordinates. The input 3D model of ligand SAH for docking was extracted from the PDB entry 2EGW and used again as an input file for MD [24,25,41,42,43,44,45]. The *N. gonorrhoeae* RsmE monomer topology and its coordinates were generated using Gromacs 2020.4 tools [46,47,48,49,50]. The molecular system was set to electroneutrality (zero total charge) by adding counter ions with a Gromcs routine and thereupon subjected to potential energy minimization by the Steepest Descent algorithm and then equilibrated by NVT (V-rescale, modified Berendsen thermostat) and NPT for 0.1 ns at 300 K each. The RsmE monomers of *N. gonorrhoeae* alone and in the presence of SAH were simulated by MD on the center of a dodecahedron box filled with SPCE water for 100 ns (50,000,000 steps, dt = 0.002 ps) under the Verlet scheme and Parrinello–Rahman barostat at 300 K in an NPT assembly, applying the most recent release of CHARMM36 all-atom force field with Gromacs 2020.4 [24,25,42,43,44,45,46,47,48,49,50]. Time-step atom trajectories and SAH-RsmE interaction distances were analyzed using the VMD package version 1.9.3 [51]. Plots were generated using Grace, version 5.1.19 [52].

### 4.5. Model Limitations and Study Design Implications 

Running molecular dynamics or docking programs becomes a daunting task even during routine work with default values of standard settings because the bottleneck has always been the small number of predefined atoms along with their bonds, partial charges and other physicochemical propensities. Straightforward adding new parameters to the built-in biological building blocks—like amino acids for proteins and nucleotides for RNA or DNA—is hampered by the need for calibration and validation within the framework of that force field. Newcomers or inexperienced novices tend to suppress or omit aspects concerning model limitations and the consequences thereof for the study design, fearing their publication could be rejected. This way most valuable information for improving forthcoming program releases or the mere chance for other researchers to overcome the detected downsides will be lost.

During our MD study, we used the force field package FF CHARMM36. We report that the methylated sulfur atom—having a total net charge of "+1"—is not recognized by the MD tool CGenFF. The flaw is a more general downside of the underlying ChARMM force field in that any ligand with positive net charges (+1, +2, etc.) cannot be treated in a straightforward way due to missing parametrization. Our case fits into this context because ligand SAM has a sulfur atom which is positively charged due to its additional methyl group. To our best knowledge, in none of the standard versions of MD force field packages (CHARMM, GROMOS, NAMD) is there a validated parametrization for a positively charged S atom. As a direct consequence without possibilities for validated parameter input, no SAM topology file can be created. Over time, solutions have been discussed on Web-based fora (forums). So far, no CHARMM36-validated or all-purpose workarounds have been presented to the MD community. Hence, the conundrum of missing parameters still continues.

The degree of ionization depends on the surrounding solvent and the electronic influence of the substituents on the solute. In the general case of carboxyl group dissociation into carboxylate anions on the side chains, the influence of the scaffold substituents is greatly reduced if the solvent assists the proton in leaving the carboxyl group. In our case, water constitutes this proton donating and accepting polar solvent. It shows strong ionizing power for the amine and carboxyl group on ligand SAM or SAH. In addition, water does not only ease the dissociation, but also stabilizes both the cationic form of the amine group along with its leaving anionic hydroxyl moiety. In the case of the ligand´s organic acid group water stabilizes its corresponding anionic form along with its leaving group, a cationic proton (by solvation/hydration). In a lipophilic pocket—which is the case here—it is extremely unlikely to encounter both groups in their dissociated forms (Figure 15). Without ionization, the far more non-polar neutral forms of amine and carboxyl groups on our two ligands will prevail. As an approximation, we ran MD and docking simulations with the neutral form of the methionyl head group. Physicochemical in vitro studies revealed that it is about 10^15^ times as difficult to dissociate a carboxyl group in a nonpolar organic solvent as in water (Figure 15).

Moreover, upon inspection and docking analyses, it was concluded that all 64 water molecules belong to the crystal packaging during crystallo-genesis. In particular, the depicted water moiety (Figure 15) on chain B (HOH 411) does not belong to the elucidated biological unit. Any solvent molecules would fit and participate in elaborated hydrogen bond networking. The entrance to the deep binding cleft has a mixed lipophilic/hydrophilic front end (outer lip) and an outspokenly hydrophobic back wall (Figure 15). Consistently, none of our cited literature in the field of structural biology reports on any water-mediated interaction with the elucidated catalytic mechanism. 

To circumvent these general problems about non-validated CHARMM36 force field patches we combined MD with complementary approaches. This way, as a valuable asset, our study not only relies on validated CHARMM36 FF settings but also on atomic scale insight from the ligand docking of SAM and SAH at the binding site in addition to structure-bases analysis of interacting amino acids based on evidence from the literature. The atomic scale insight was complemented by structural comparison with related PDB entries in Appendix A of the Appendix A [21,22,53,54,55,56,57,58,59]. Details regarding protonation states of SAM and SAH [60,61] as well as the evaluation of docking energies are provided in Appendix A [62,63,64,65]. 

## 5. Conclusions

The study succeeded in combining docking and molecular dynamics simulations to gain complementary insight into the theoretical mechanism of SAH as a leaving group at the atomic scale. Taken together all findings, the active conformations of SAM and SAH in the crystal structures and molecular docking showed very similar geometries (torsions) in practically the same poses. Comparison revealed conserved interaction patterns for *N. gonorrhoeae* and *A. aeolicus*, with little differences from *E. coli*. Likewise, the binding energies for SAM and SAH in *N. gonorrhoeae* and *A. aeolicus* were highly similar, hinting at a general catalytic mechanism for bacteria. In addition to the crystal structures and docking results, molecular dynamics simulations of *N. gonorrhoeae* helped identify pivotal interacting residues, namely Leu215, Arg218 and Leu220. They sit on the connecting loop segment between β12 and α6, along with Met169. They are either conserved or undergo homologous exchange and specifically orchestrate not only SAM binding but also SAH release upon transmethylation. Homologous residue exchange was based on the identification of equivalent positions in *A. aeloicus* and *E. coli*. It is safe to say that they represent a more general molecular mechanism and thereby the conserved enzyme catalytic function.

The energy minimization of *E. coli* RsmE by MD allowed us to obtain poses very similar to those of the crystal structures of the other two species. The binding energies in *E. coli* were close to those estimated for *N. gonorrhoeae* and *A. aeolicus*. However, it was not possible to reproduce the interaction of the hydroxyl O2’ of SAM and SAH with Gly192 and Gly185 in *N. gonorrhoeae* and *A. aeolicus* RsmE, respectively.

Initial docking results with the crystal structure from *E. coli* RsmE showed large differences derived from the different shapes and large sizes of the binding sites compared to the others. It was found to be an open or exposed cavity that adjusts to the size of the ligand once it binds. The binding site was found to be in a widely open state. In the presence of SAM, the binding site switches to a narrowly closed state. On the other hand, the MD simulation in the presence of SAH allowed us to distinguish the alternating interaction of the SAH carboxyl group with Arg221 and Thr222, two residues that contribute decisively to ligand release.

MD suggests that the weakening of the interaction between the SAH hydroxyl groups and Gly192 is critical for promoting the opening of the gate formed by residues Ile171 and Ile219 in *N. gonorrhoeae*. Further experimentation with selective mutagenesis will be the key to demonstrating this role. The binding site opens when the distance between the two isoleucine residues is greater than 6 Å. It is expected to be exposed to the neighboring water molecules, which were not investigated as to whether they comply with a structural role or interaction since the RMSD and Rg parameters indicate that the binding site undergoes significant shape variations. However, there is still a lack of information to model a ternary complex between the p44 helix, SAM and RsmE, and to gather a complete catalytic mechanism that may be employed to design a new generation of antibiotics to improve the clinical treatment of a wide spectrum of drug-resistant bacteria.

## Figures and Tables

**Figure 1 ijms-24-16722-f001:**
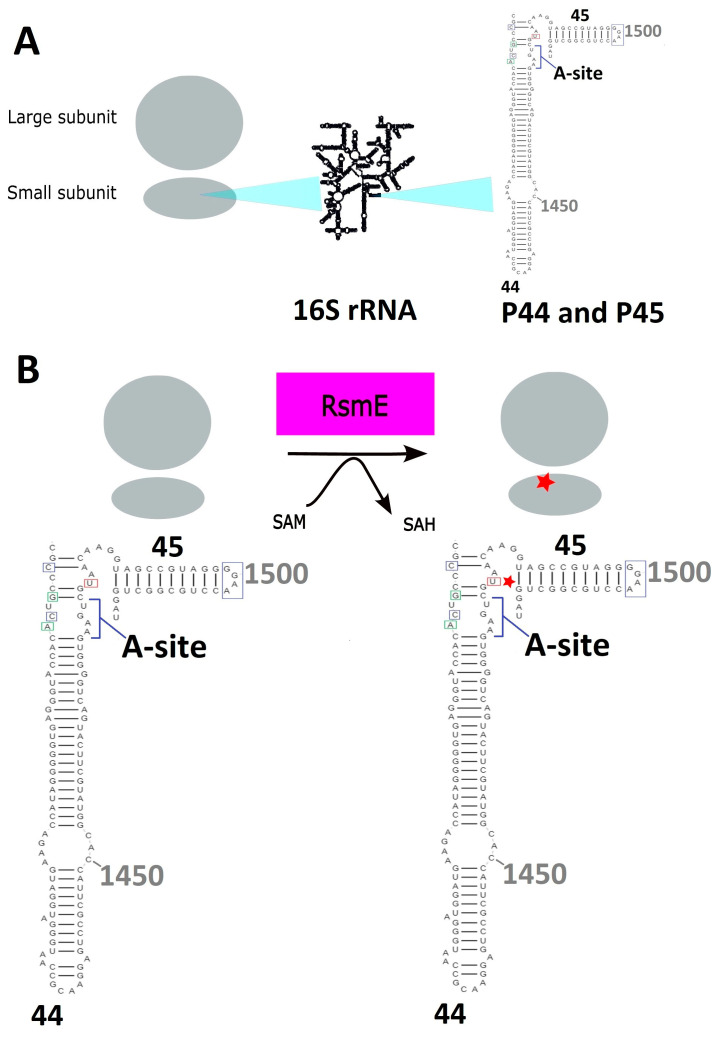
(**A**) P44 and P45 are part of the large RNA-protein complex that ensembles the small subunit of ribosomes in bacteria. (**B**) The P44 and P45 contain the A-site where RsmE methylates the uridine at position 1498 which is in proximity to other sites that are methylated (enclosed in green boxes) by other methyltransferases. RsmE forms a holoenzyme with S-Adenosylmethionine as a cofactor in order to transfer a methyl group. The star (red asterisk) depicts the position of uridine methylation (red box) in the ribosome for protein biosynthesis.

**Figure 2 ijms-24-16722-f002:**
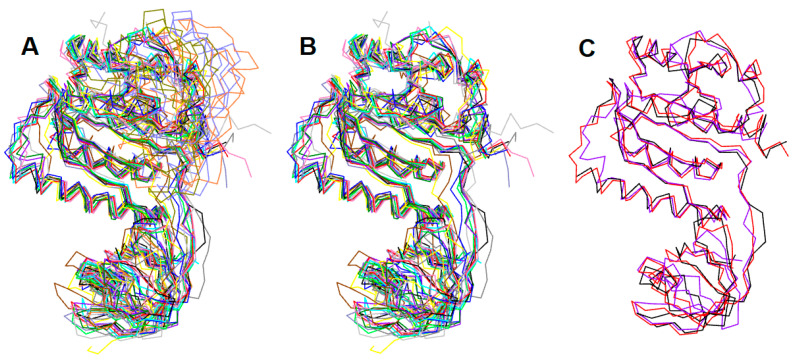
(**A**) Superposition of sixteen RsmE crystal structures of the following organisms: *Haemophilus influenzae* (1NXZ, 1VHY)*, Thermus thermophilus* (1V6Z,2CX8,2Z0Y), *Bacillus subtilis* (1VHK), *Thermotoga maritima* (1Z85), *Thermus thermophilus* (1V6Z, 2CX8, 2Z0Y), *Aquifex aeolicus* (2EGV, 2EGW), *Porphyromonas gingivalis* (3KW2), *E. coli* (4E8B), *Sinorhizobium meliloti* (4J3C), *Mycobacterium tuberculosis* (4L69), *Legionella pneumophila* (5O95, 5O96) and *Neisseria gonorrhoeae* (5VM8). (**B**) Crystal structures superposition of the RsmE monomers without *Thermus thermophilus* species. (**C**) *Neisseria gonorrhoeae* (black), *E. coli* (red) and *Aquifex aeolicus* (purple) RsmE superposition (5VM8, 4E8B, 2EGW, respectively). For details and color codes, see Appendix A.

**Figure 3 ijms-24-16722-f003:**
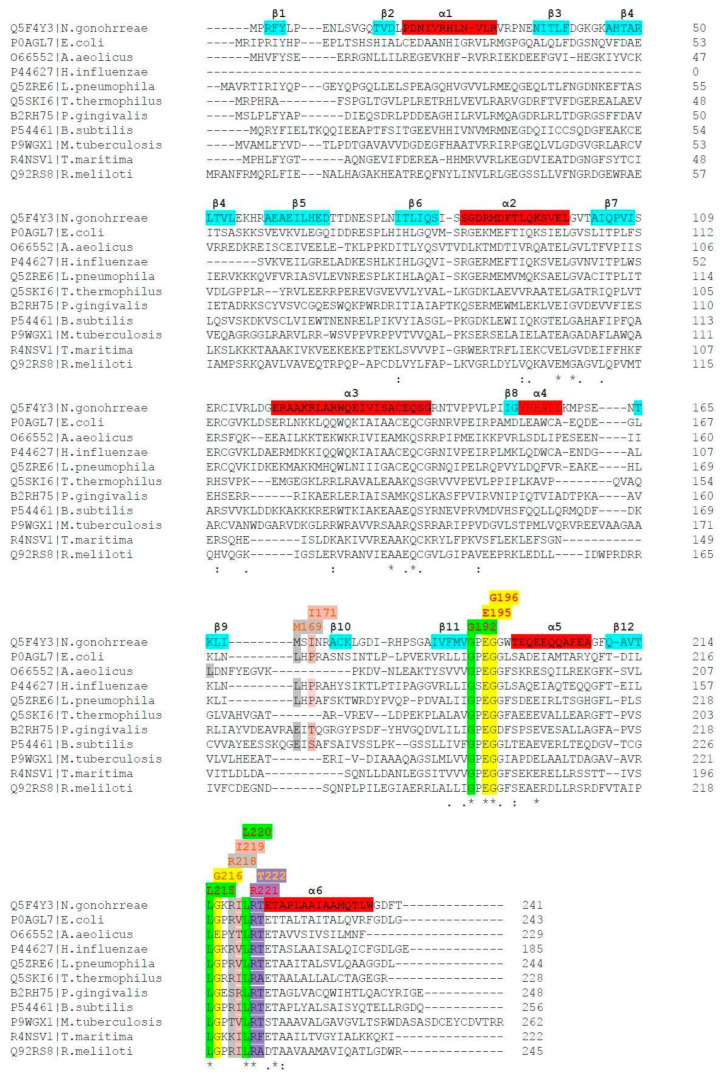
Multiple sequence alignment (MSA) of all available Mtases with elucidated structures. The last column labels the identity number of the last amino acid (in one-letter code) in each row; the last row of each alignment block indicates either residue identity by an asterisk ( * ) or homology with strongly similar properties by a colon ( : ). A period ( . ) indicates poor conservation between amino acids with weakly similar properties in addition to a blank space ( ) in this line which symbolizes that one or more amino acids in this column are not conserved. For background color coding, cf. text below.

**Figure 4 ijms-24-16722-f004:**
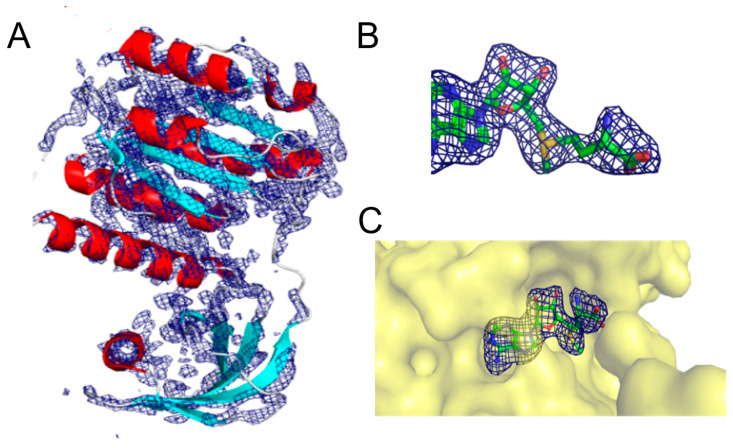
(**A**) Superposition of *N. gonorrhoeae* RsmE (PDB entry: 5VM8) on its density map. (**B**) Superposition of the Polder map on the SAM model. (**C**) RsmE binding site in complex with SAM. RsmE is represented as a surface. In all panels, the electron density map and Polder map are represented as a blue meshwork. The density and Polder maps are contoured at 2.5σ. Polder and 2mF0-Fc maps were processed with PHENIX using default settings and further prepared with PYMOL. Of note, a Polder map is a more suitable omit map for ligands to exclude possible bulk solvent that obscures weak densities that may not be visible otherwise.

**Figure 5 ijms-24-16722-f005:**
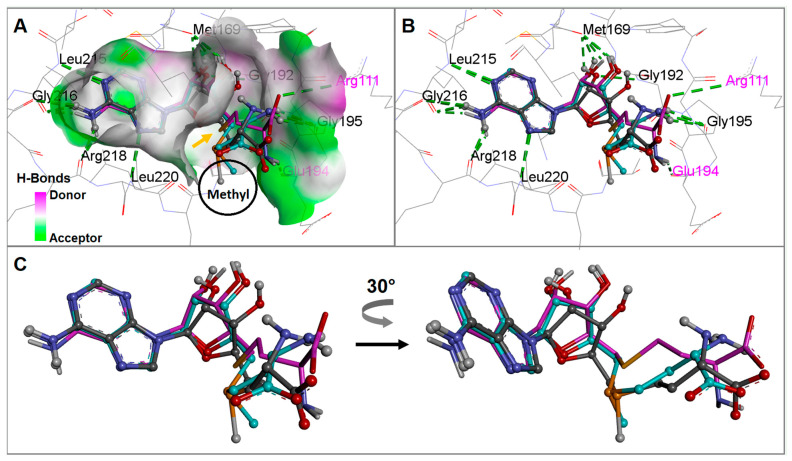
Conformational analysis of experimentally determined poses from crystal complexes (EDP) of SAM *versus* final docking solutions (FDS) of SAM or SAH. Of note, docked solutions for SAM and SAH also complement the limited crystal structures at hand. The conformational changes are displayed at the RsmE binding site of *N. gonorrhoeae*. (**A**) the surface of the RsmE binding site is shaded by hydrogen-bonding potential. The methyl positions of SAM in EDP or FDS are circled and labeled “Methyl” to indicate how the sulfur atom position in SAH lies further inwards than in SAM in EDP or FDS (orange arrow). Residue labels are only for positional reference. (**B**) H-bonds and hydrophobic interactions of SAM or SAH with residues at the binding site of *N. gonorrhoeae* RsmE are displayed by green dotted lines. The magenta-colored residues show interaction with the homocysteine tail of docked SAH. (**C**) Superposition of the extracted active conformations for SAM in EDP together with SAM and SAH, both in FDS. Style and color coding for the three ligand models: (i) crystal SAM—ball and sticks with gray carbon atoms; (ii) docked SAM—ball and sticks with light blue carbon atoms; and (iii) docked SAH—sticks with magenta-colored carbon atoms.

**Figure 6 ijms-24-16722-f006:**
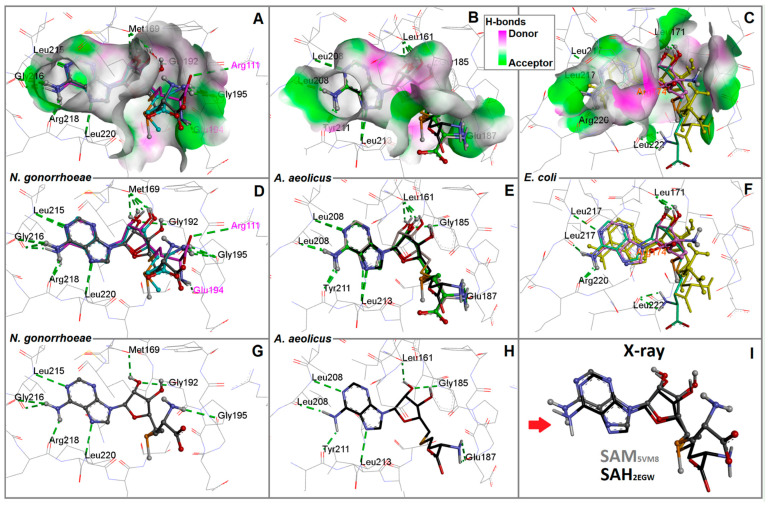
Superposition analysis of experimentally determined poses from crystal complexes (EDP) *versus* final docking solutions (FDS). Panels **A**–**C** show EDP of SAM and SAH at the RsmE binding sites of *N. gonorrhoeae, A. aeolicus* and *E. coli* in, respectively. The inlay shows the surface color code for H-bonding. Balls and stick models for SAM or SAH as stick models. (**D**) SAM in EDP, SAM (light blue carbon atoms) and SAH (magenta carbons) in FDS. Hydrogen bonds are displayed within the RsmE binding site of *N. gonorrhoeae*. (**E**) SAH in EDP, SAH in FDS (light gray carbon atoms) and SAM in FDS (green carbons) in interaction with residues by H-bonds at the RsmE binding site of *A. aeolicus*. (**F**) SAM and SAH in EDP (yellow), along with SAM (violet carbons) and SAH (metallic green colored carbon atoms) in FDS. (**G**) SAM in EDP (gray carbon atoms) interacting with residues by H-bonds at the RsmE binding site of *N. gonorrhoeae* (5VM8). (**H**) SAH in EDP (black carbon atoms) interacts with residues by H-bonds at the RsmE binding site of *A. aeolicus* (2EGW). (**I**) Superpositions of SAM and SAH in EDP which were extracted from their *N. gonorrhoeae* and *A. aeolicus* RsmEs complexes (PDB codes: 5VM8 and 2EGW), respectively in panels (**G**,**H**) (red arrow).

**Figure 7 ijms-24-16722-f007:**
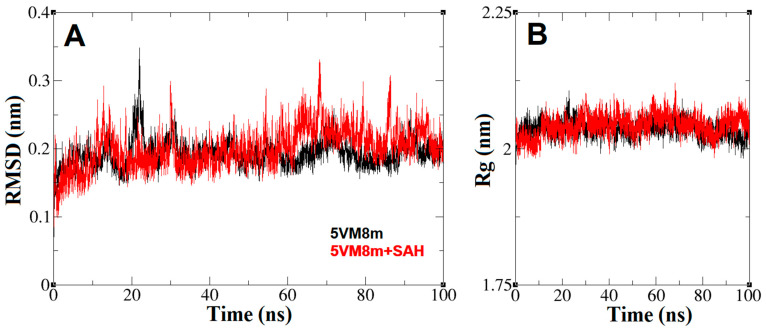
(**A**) Root mean square deviation or RMSD (nm) vs. MD simulation time (ns) plots. Mean values: RSMD_5VM8m_ = 2.73 Å, RSMD_5VM8m+SAH_ = 2.89 Å. (**B**) Radius of gyration or Rg (nm) vs. MD simulation time (ns) plots. Mean values: Rg_5VM8m_ = 2.0369 nm, Rg_5VM8m+SAH_ = 2.0427 nm. In both plots, the black line represents the MD simulation only with the RsmE monomer alone (5VM8m) and the red line represents the MD simulation of the RsmE monomer in the presence of SAH (5VM8m + SAH). Both are calculated over backbone atoms.

**Figure 8 ijms-24-16722-f008:**
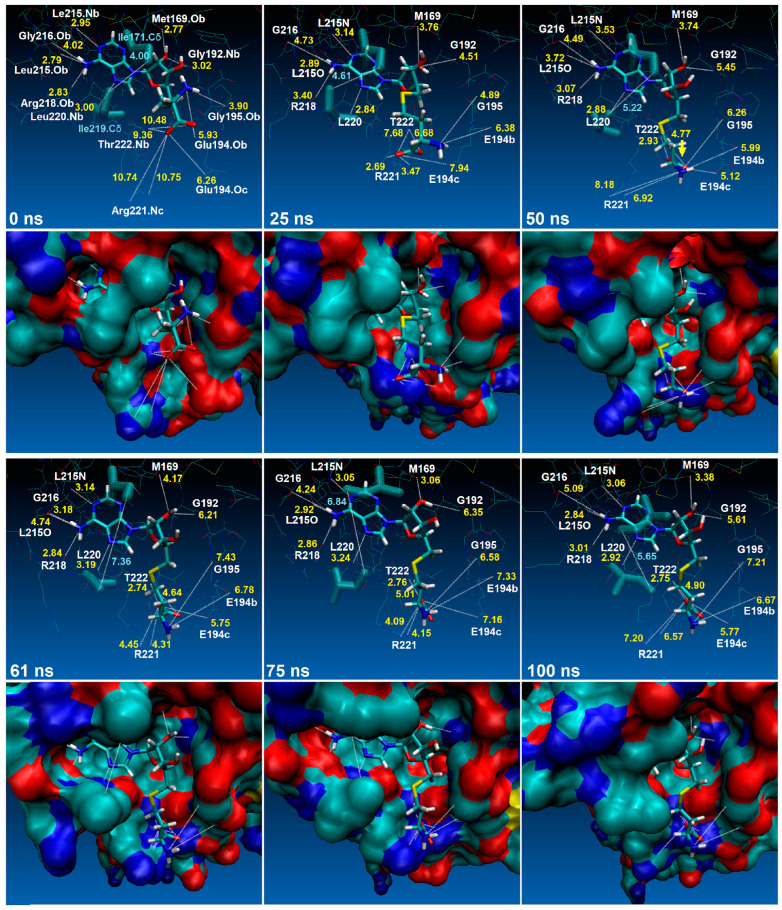
Time steps of *N. gonorrhoeae* RsmE and SAH interaction distances observed by MD simulation. For each time step, the RsmE residues are represented in line models (1st and 3rd rows) and solid backbone surface (2nd and 4th rows). Measured distance values are expressed in Å. The distances between Ile171.Cδ and Ile219.Cδ (translucent sticks) are colored in light blue. All other distances are colored in yellow. Only the first image (0 ns, upper left) shows details about the receptor residue atoms interacting with SAH. In the rest of the images, the one-letter notation is used to designate each residue. Side-chain atoms are designated with letter “c” and backbone atoms with letter “b” at the end of each label. The yellow arrow indicates the direction of SAH´s exit mechanism (cf. distance changes).

**Figure 9 ijms-24-16722-f009:**
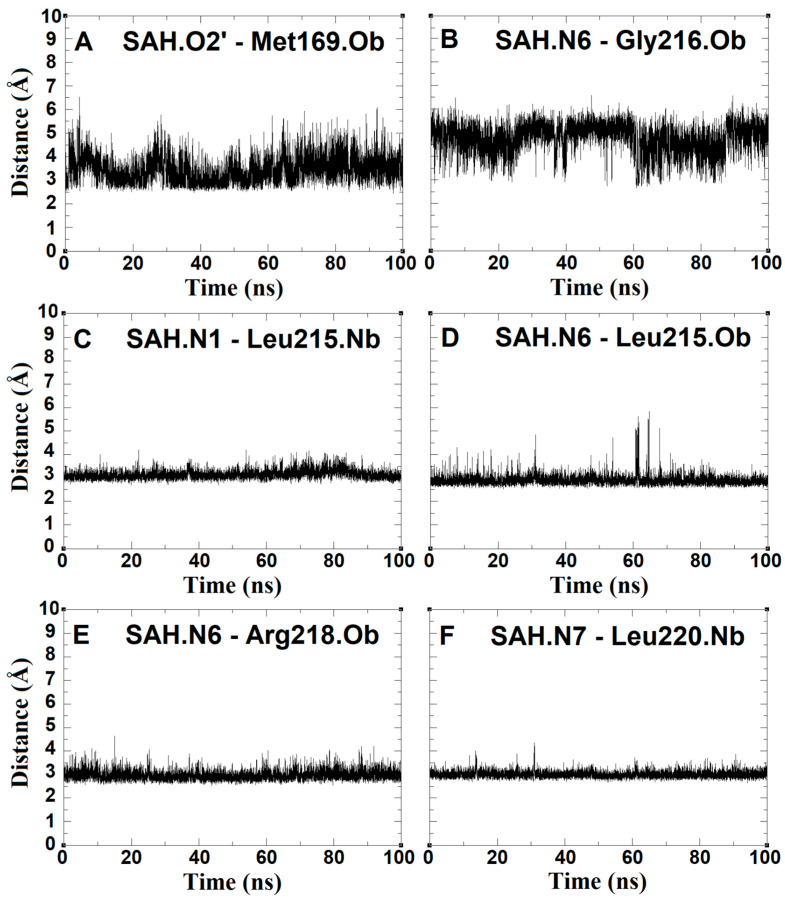
Distance (Å) vs. time (ns) plots between the adenine and its sugar ring atoms of SAH and the atoms of the *N. gonohrroeae* RsmE binding site observed during the molecular dynamics. Side chain atoms are designated with letter “c” and backbone atoms with letter “b” at the end of each label. The scale is exactly the same for all charts. Panels (**A**–**F)** disclose the movements over time (100 nanoseconds) in a pairwise manner for mechanistically relevant molecule parts. For instance, panel (**F**) documents the distance fluctuation between nitrogen atom labeled 7 and the backbone nitrogen (Nb) of leucine 220.

**Figure 10 ijms-24-16722-f010:**
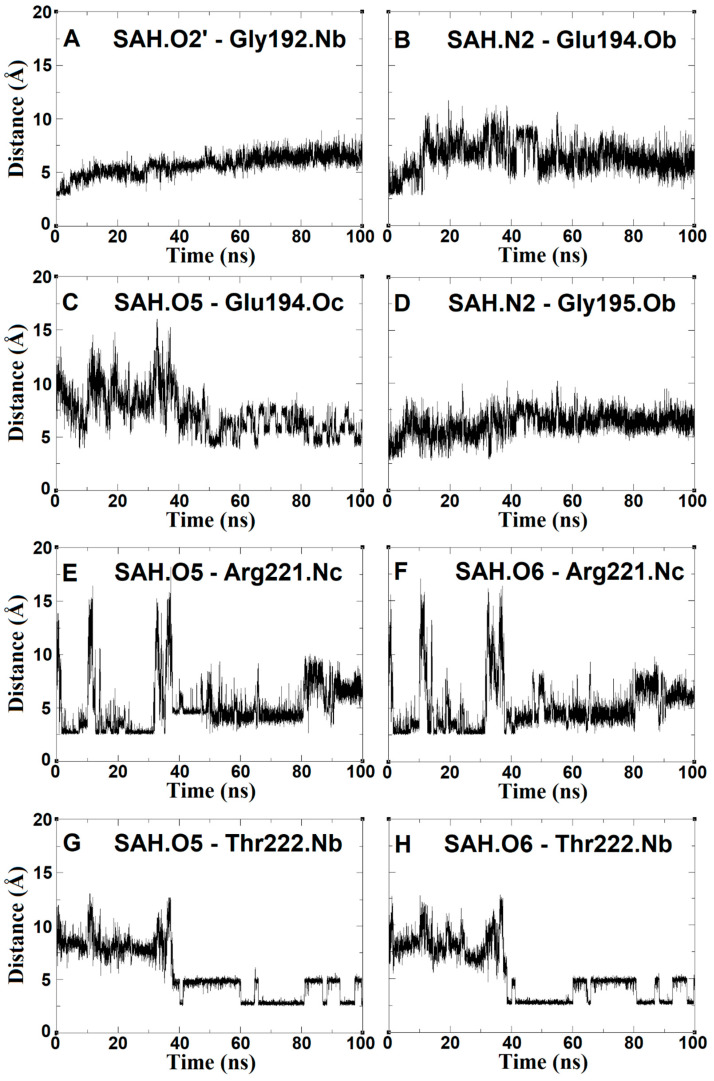
Distance (Å) vs. time (ns) plots between the homocysteine tail atoms of SAH and the residue atoms of the *N. gonohrroeae* RsmE binding site observed during molecular dynamics. Side-chain atoms are designated with letter “c” and backbone atoms with letter “b”, at the end of each label. The scale is exactly the same for all charts.

**Figure 11 ijms-24-16722-f011:**
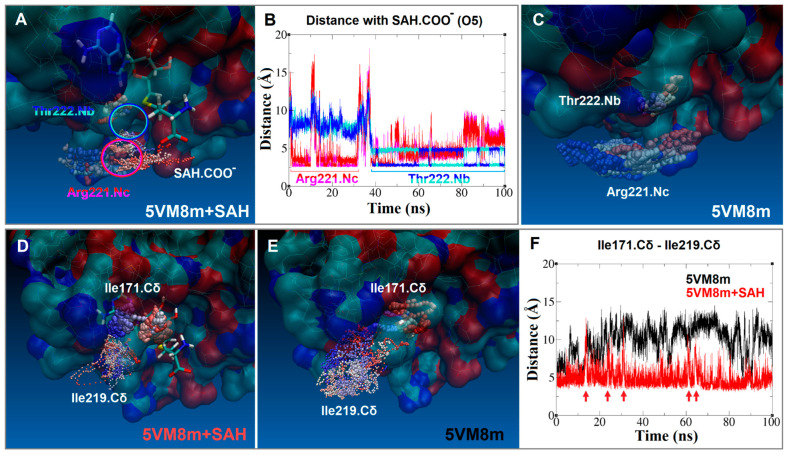
(**A**) Trajectories of SAH carboxyl group (SAH.COO^–^, small solid balls), Arg221 side chain nitrogen (Arg221.Nc, translucent balls) and Thr222 backbone nitrogen (Thr222.Nb, solid balls). (**B**) Distance (Å) vs. time (ns) plot of the SAH carboxyl group with atoms of residues Arg221 and Thr222 (after merging plots E, F, G, H of Figure 10). Color code: fluctuations in red (Arg221.Nc) or blue (Thr222.Nb). (**C**) Trajectories of Arg221 side chain nitrogen (Arg221.Nc, translucent balls) and Thr222 backbone nitrogen (Thr222.Nb, solid balls) tracked in the MD simulation with the RsmE monomer alone. (**D**) Trajectories of Ile171 side-chain delta carbon (Ile171.Cδ, solid balls) and Ile219 side-chain delta carbon (Ile219.Cδ, solid small balls) tracked in the RsmE monomer and SAH MD. (**E**) Trajectories of Ile171 side-chain delta carbon (Ile171.Cδ, solid balls) and Ile219 side-chain delta carbon (Ile219.Cδ, solid small balls) tracked in the MD with only the RsmE monomer. (**F**) Distance (Å) vs. time (ns) plot between the Ile171 and Ile219 side chains delta carbons. The red arrows indicate maximum points on the red line that correspond to the time instances when the binding site is completely open. All trajectories were aligned with the VMD Trajectory tool and were tracked every 5 steps (over 10,000). Color code for time step trajectories: red trajectory steps are in the first third of simulation time (0 to 33 ns approx.), white trajectory steps belong to the second third of simulation time (33 to 66 ns approx.) and blue trajectory steps are at the end of simulation time (66 to 100 ns approx.). All the trajectories are superimposed with the receptor in translucent surface representation and ligand represented in sticks at time step 0.

**Figure 12 ijms-24-16722-f012:**
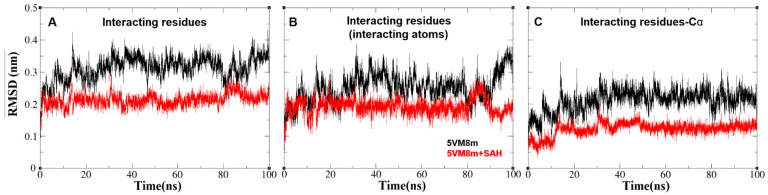
Root mean square distance or RMSD (nm) vs. MD simulation time (ns) plots of the *N. gonorrhoeae* RsmE binding site of the interacting residues and its atoms of Met169.Ob, Ile171.Cδ, Gly192.Nb, Glu194.(Ob, Oc), Gly195.Ob, Leu215.(Nb, Ob), Gly216.Ob, Arg218Ob, Ile219.Cδ, Leu220.Nb, Arg221.Nc and Thr222.Nb. (**A**) RMSD plotted over all the residue atoms. (**B**) RMSD plotted over all the interacting residue atoms. (**C**) RMSD plotted over all the residue Cα atoms. The scale is the same for all plots.

**Figure 13 ijms-24-16722-f013:**
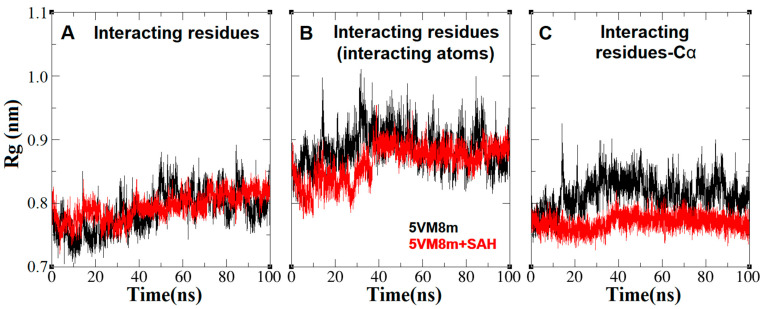
Radius of gyration (nm) vs. MD simulation time (ns) plots of the *N. gonorrhoeae* RsmE binding site of the interacting residues and its atoms of Met169.Ob, Ile171.Cδ, Gly192.Nb, Glu194.(Ob, Oc), Gly195.Ob, Leu215.(Nb, Ob), Gly216.Ob, Arg218Ob, Ile219.Cδ, Leu220.Nb, Arg221.Nc and Thr222.Nb. (**A**) Rg plotted over all the residue atoms. (**B**) Rg plotted over all the interacting residue atoms. (**C**) Rg plotted over all the residues Cα atoms. The scale is the same for all plots.

**Figure 14 ijms-24-16722-f014:**
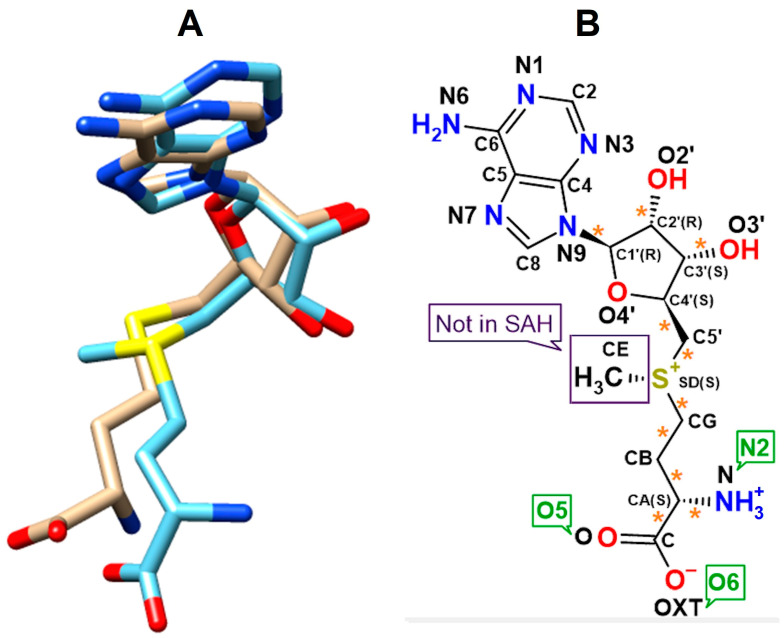
(**A**) Superposition of S-adenosyl-L-methionine (SAM, blue) and S-Adenosyl-L-homocysteine (SAH, beige). All hydrogen atoms were omitted for visibility of hetero atom alignment. (**B**) Atom numbering for monocationic SAM and neutral SAH (labeled “Not in SAH”). The twitter-ionic form of the methionine head group is displayed. The conventional notation is given by black letters next to the corresponding atom. Here, labels (green) were reassigned to make it easier to refer to certain atoms. Adapted from published liganded complexes of ribosomal RNA small subunit methyltransferase from *N. gonorrhoeae* with SAM (PDB code: 5VM8), and ribosomal RNA methyltransferase with SAH from *A. aeolicus* (PDB code: 2EGW). The asterisks mark rotatable bonds under Autodock 4.2. Chiral centers are labeled by (S or R).

**Figure 15 ijms-24-16722-f015:**
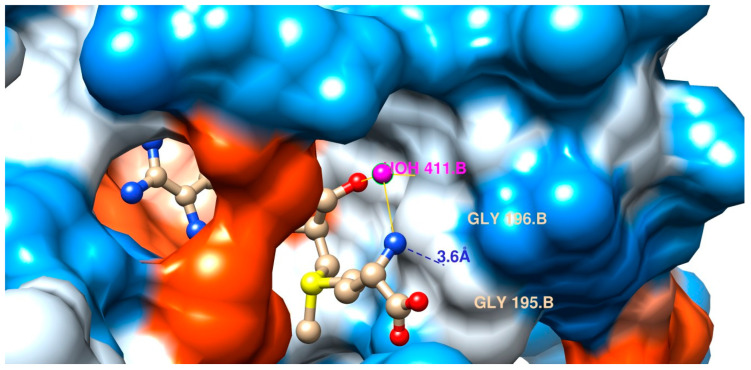
Top-down view on the entrance of the SAM-liganded binding site with the hydrophilic (bluish tones) and lipophilic (reddish tones) protein surface. The entrance has a wide opening for the natural substrate in the foreground (bottom right side). The binding cleft is occupied by SAM (atom-colored ball and stick model). The edge of entrance is displayed as a white spot to symbolize water-filled space (rightmost and bottommost borders). The entrance has an outspokenly lipophilic outer lip to the viewing front (red, mid-section) in addition to a mixed back wall (to the right side behind the ligand). The mix is due to a hydrophilic outward lip (blue) and a neutral inward lip (grey/white). The entire inside surface of the binding cleft is hydrophobic in nature (red, inwards). Only the sharply limited “L”-formed outer lip is outspokenly hydrophilic (blue, top to bottom right). The terminal amino acid side chain of the methionyl substructure of SAM is located in a sandwiched position between lipophilic and hydrophilic outer lips of the entrance (bottommost, central part).

## Data Availability

Data are contained within the article and Appendix A.

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
