# Peer review of "Molecular Dynamics and Docking Simulations of Homologous RsmE Methyltransferases Hints at a General Mechanism for Substrate Release upon Uridine Methylation on 16S rRNA"

_ijms, 2023, doi:10.3390/ijms242316722_

Round 1
Reviewer 1 Report
Comments and Suggestions for Authors
The authors presented the manuscript entitled: “Molecular dynamics and docking simulations of homologous RsmE methyltransferases hints at a general mechanism for substrate release upon uridine methylation on 16S rRNA”. In this manuscript, the authors applied molecular dynamics and docking simulations on the crystal structure of Neisseria Gonorrhoeae RsmE compared to Escherichia coli and Aquifex aeolicus. The authors proposed for the first time a general catalytic mechanism for bacterial RsmE and a computational approach to analyze the exit mechanism of S-adenosylhomocysteine by molecular dynamics simulations.
In my opinion, this is a valuable work and will be an interesting source for the scientific community.
I suggest the following corrections:
In some figures amino acids and numbers overlap and therefore can’t be seen clearly. Please revise and modify.
Use the mdpi rules for the references. Any reference has a DOI.
Author Response
We accept all comments with gratitude and join a PDF with the details for the reviewer 1.

Reviewer 2 Report
Comments and Suggestions for Authors
1) RsmE is a methyltransferase (MTase) and was first discovered in Escherichia coli
Provide appropriate reference after above line.
2) Compared to this N-methylation, C-methylation 88 requires a totally different mechanism since the Arrhenius (or activation potential) energy 89 barrier lies much higher than for m3U1498.
Justify and explain the above fact about barrier energy.
3) A deeper understanding of the biochemical steps during the catalytic methylation 91 cycle of RsmE is still lacking. Hence, we propose a computational approach to analyze the 92 exit mechanism of SAH by molecular dynamics simulations.
Provide appropriate reference for MD simulation to justify its use and significance. Ref like:
Wasif Baig, M., Pederzoli, M., Kývala, M., Cwiklik, L., & Pittner, J. (2021). Theoretical investigation of the effect of alkylation and bromination on intersystem crossing in BODIPY-based photosensitizers. The Journal of Physical Chemistry B, 125(42), 11617-11627.
4) To further investigate the catalytic mechanism of RsmE, the structural model of 5VM8 was taken as a reference because it yielded the most consistent results and unveiled the most continuous electron density map.
Kindly provide details after above text on Page 6 how electron density were obtained.
5) The 194 carboxyl group of SAM faces towards the binding site exit without making any other 195 contact with RsmE, whereas its amino group is attracted by the backbone oxygen of 196 Gly195.
Please mention here whether hydrogen bonding is present between these groups or not.
6) This interaction dissimilarity between SAM and SAH side 228 chains is due to the inward displacement of the sulfur atom that allows the methyl group 229 to swing over during the ongoing methyl transfer process Compute free energy changes for displacement of sulfur atom and swinging motion of methyl group.
7) The distance between the SAH adenine ring and the amine group with Leu215, 348 Arg218, and Leu220 residues remained practically stable at 3 Å for the entire simulation
Please explain after above text that how stability is inferred from distance of 3 Angstroms. Justify and explain this fact please.
8) In contrast, however, when it is present, its side 377 chain terminal carboxyl group adopts an orderly and defined back and forth rotation 378 similar to a swing. It should be more appropriate if you can compute free energy changes for above mentioned conformational changes.
9) while the adenosyl side chain remains 477 unprotonated for an estimated 99 to 99.9 % (pka 4 – pH 7 = 99.9% free base).
Fix the typo in above sentence at page 9. It should be pKa not pka.
Author Response
We accept all comments and carried out all modifications as requested to improve our work. Thank you for the corrections and care-taking of our work.
